# Inovirus-Encoded Peptides Induce Specific Toxicity in *Pseudomonas aeruginosa*

**DOI:** 10.3390/v17010112

**Published:** 2025-01-15

**Authors:** Juehua Weng, Yunxue Guo, Jiayu Gu, Ran Chen, Xiaoxue Wang

**Affiliations:** 1Key Laboratory of Tropical Marine Bio-resources and Ecology, South China Sea Institute of Oceanology, Chinese Academy of Sciences, Guangzhou 511458, China; wengjuehua22@mails.ucas.ac.cn (J.W.); gujiayu20@mails.ucas.ac.cn (J.G.); chenran@scsio.ac.cn (R.C.); 2University of Chinese Academy of Sciences, Beijing 100049, China; 3Key Laboratory of Tropical Oceanography, South China Sea Institute of Oceanology, Chinese Academy of Sciences, Guangzhou 511458, China

**Keywords:** *Pseudomonas aeruginosa*, Pf filamentous phage, antimicrobial polypeptide, prophage

## Abstract

*Pseudomonas aeruginosa* is a common opportunistic pathogen associated with nosocomial infections. The primary treatment for infections typically involves antibiotics, which can lead to the emergence of multidrug-resistant strains. Therefore, there is a pressing need for safe and effective alternative methods. Phage therapy stands out as a promising approach. However, filamentous prophages (Pfs) commonly found in *P. aeruginosa* encode genes with phage defense activity, thereby reducing the efficacy of phage therapy. Through a genomic analysis of the Pf4 prophage, we identified a 102 bp gene co-transcribed with the upstream gene responsible for phage release (*zot* gene), giving rise to a 33-amino-acid polypeptide that we have named Pf4-encoded toxic polypeptide (PftP4). The overexpression of PftP4 demonstrated cellular toxicity in *P. aeruginosa*, with subcellular localization indicating its presence in the cell membrane and a subsequent increase in membrane permeability. Notably, PftP4 homologues are found in multiple Pf phages and exhibit specificity in their toxicity towards *P. aeruginosa* among the tested bacterial strains. Our study reveals that the novel Pf-encoded polypeptide PftP4 has the potential to selectively target and eradicate *P. aeruginosa*, offering valuable insights for combating *P. aeruginosa* infections.

## 1. Introduction

*Pseudomonas aeruginosa*, a Gram-negative bacterium, is widely distributed in diverse natural habitats and serves as a global opportunistic pathogen known for its capacity to provoke a broad spectrum of infections in healthcare settings, notably leading to morbidity and mortality in cystic fibrosis patients and immunocompromised individuals [1]. Within the lungs of cystic fibrosis patients, *P. aeruginosa* tends to establish resilient biofilms, complicating therapeutic interventions [2]. In addition, this bacterium is notorious for its resistance to multiple antibiotics, rendering infections challenging to manage and resulting in prolonged hospitalization and heightened mortality rates. The impact of *P. aeruginosa* infections transcends individual cases, as they also pose a significant burden on healthcare systems and resources. Data from the China Antimicrobial Surveillance Network (CHINET) (http://www.chinets.com/) (accessed on 12 December 2024) showed that *P. aeruginosa* ranked as the fifth most common nosocomial pathogen, constituting 7.8% of the 445,199 clinically isolated pathogenic strains. A comprehensive grasp of the pathogenic mechanisms of *P. aeruginosa* is imperative for devising effective strategies to combat its infections.

In recent years, the treatment of *P. aeruginosa* infections has witnessed the development and applications of several innovative strategies aimed at tackling the challenges posed by this resilient pathogen. One prominent approach involves utilizing combination therapies, which harness the synergistic effects of multiple antimicrobial agents to combat *P. aeruginosa* infections and reduce the risk of resistance development. Furthermore, advancements in precision medicine have spurred on the investigation of personalized treatment regimens customized to the specific genetic and phenotypic traits of both the pathogen and the infected individual. While antibiotics remain the primary therapeutic option for *P. aeruginosa* infections, the re-emergence of the multi-drug resistance of *P. aeruginosa* pathogens presents a significant challenge. Recent research has highlighted various non-antibiotic therapeutic strategies that effectively target *P. aeruginosa*, including phage therapy [3,4], vaccine therapeutics [5], antibody–drug conjugates [6], nanoparticles [7], antimicrobial peptides [8,9], quorum sensing inhibitors [10,11], iron chelators [12], and electrochemical scaffolds [13]. Additionally, incorporating immunotherapeutic strategies and developing vaccines that target *P. aeruginosa*’s virulence factors offer promising avenues for enhancing the host’s immune response and preventing recurrent infections. These multifaceted and continuously evolving strategies highlight the dynamic landscape of *P. aeruginosa* infection treatment, demonstrating a concerted endeavor to combat this formidable pathogen with a diverse array of therapeutic options.

A distinguishing characteristic of *P. aeruginosa* is the prevalent integration of its chromosome with Pf filamentous phages [14,15]. Pf phages, belonging to *Inoviruses*, replicate within their hosts in a temperate lifestyle as prophages and are excised from the host chromosome when induced into replicative form [16]. These phages co-evolve with their hosts, and significantly influence various aspects such as biofilm formation [17], virulence [18], quorum sensing [16,18], phage defense [19,20], antibiotic resistance [15], and pathogenesis in mammals [21,22] of *P. aeruginosa*. Emerging evidence suggests that Pf4, a notable filamentous phage, carries several genes, both structural and non-structural, with cellular toxicity implications. For instance, the structure gene *PA0724* encodes PIII, a phage coat protein of the Pf4 phage [19,20,23], while non-structural genes such as *pfiT* encode toxin components of a type II toxin–antitoxin system *pfiTA* [24]; *PA0726* encodes a putative Zona occludens toxin (Zot), *PA0728* encodes the integrase [25]. Furthermore, the overproduction of the toxin gene *pfkA*-*pfkB* from the type VII toxin–antitoxin system KKP (kinase–kinase–phosphatase) demonstrated strong toxicity to host cells by targeting various functional genes involved in cell division, replication, transcription, and translation processes [20]. However, the precise molecular mechanisms underlying the toxicity of these genes remain inadequately elucidated.

In this study, we focused on the Pf4-encoded polypeptide PA0726.1, which we named PftP4. We observed that PftP4 is a toxic polypeptide located in the membrane that increases membrane permeability. PftP4 is exclusive to various Pf phages, and its toxicity towards *P. aeruginosa* host cells is specific among the bacterial strains tested. Thus, comprehending the functions and toxic mechanisms of these prophage-encoded genes that induce specific toxicity in *P. aeruginosa* shows potential for uncovering novel strategies for targeted interventions and treatments.

## 2. Materials and Methods

### 2.1. Bacterial Strains and Culture Conditions

The bacterial strains and plasmids used in this study are listed in Table 1. Laboratory *P. aeruginosa* strain MPAO1, clinical *P. aeruginosa* strains, *Escherichia coli*, and *Providencia stuartii* were cultured in LB medium at 37 °C, while *Halomonas meridiana* was cultured in 2216E medium at 30 °C. Carbenicillin (100 µg/mL) was used to maintain pHERD20T-based plasmids, and chloramphenicol (15 µg/mL) was used to maintain pHGE-based plasmids. The expression of pHERD20T-based plasmids was induced with 10 mM *L*-arabinose, and the expression of pHGE-based plasmids was induced with 1 μM Isopropyl β-D-Thiogalactoside (IPTG).

### 2.2. Construction of Plasmids and Mutant Strains

The primers used in this study are listed in Table 2. For overproduction, the *pftP4* and *pftP6* genes were amplified from the MPAO1 genome through PCR, using genomic DNA as template. Then, the purified PCR products were ligated into the NcoI and HindIII pHERD20T plasmids or EcoRI and HindIII pHGE plasmids with Exnase II. Then, the recombination plasmids were transferred into bacterial hosts via conjugation. Hosts with recombination plasmids were confirmed using PCR and Sanger sequencing.

### 2.3. Prediction of Protein Structure

The transmembrane helices of the PftP4 polypeptide were predicted using TMHMM-2.0 online software (https://services.healthtech.dtu.dk/services/TMHMM-2.0/) (accessed on 20 November 2024). The protein structure of PftP4 was predicted with AlphaFold2 [30].

### 2.4. Bacterial Viability Assay

The bacterial viability of MPAO1 with expression plasmids was assessed through LIVE/DEAD™ *Bac*Light™ Bacterial Viability Kit (Thermo Fisher Scientific, Waltham, MA, USA) using microscopy. MPAO1/pHERD20T, MPAO1/pHERD20T-*pftP4*, and MPAO1/pHERD20T-*pftP6* were cultured in LB liquid medium with 100 µg/mL carbenicillin at 37 °C until the turbidity at 600 nm reached 1.0; then, they were induced using 10 mM *L*-arabinose for 3 h. A total of 1 μL SYTO™ 9 (green-fluorescent nucleic acid stain) and 1 μL propidium iodide (red-fluorescent nucleic acid stain) were added into 100 μL cultures and the mixture were incubated in the dark at room temperature for 15 min. A fluorescence microscope was used to observe bacterial cells. Cells with a compromised membrane that are dead or dying stain red, whereas cells with an intact membrane stain green.

### 2.5. Toxicity Assay

Cultures of different *P. aeruginosa*, *E. coli*, and *P. stuartii* strains with expression plasmids were grown at 37 °C overnight. Then, a loopful of the cultured samples was taken on a sterile loop, and was streaked back and forth in a zigzag pattern over part of the LB plates with 100 µg/mL carbenicillin, as well as the LB plates with 100 µg/mL carbenicillin and 10 mM *L*-arabinose. Cultures of the *H. meridiana* 43005 strain with expression plasmids were grown at 30 °C overnight. Then, a loopful of the cultured samples was taken on a sterile loop, and was streaked back and forth in a zigzag pattern over part of the 2216E plates with 15 µg/mL chloramphenicol, as well as the 2216E plates with 15 µg/mL chloramphenicol and 1 mM IPTG.

### 2.6. Subcellular Localization

To determine the localization of PftP4 in MPAO1, a *gfp* tag was fused to *pftP4* through PCR to construct MPAO1/pHERD20T-*gfp*-*pftp4*. MPAO1 with recombination pHERD20T-based plasmids were grown at 37 °C until the turbidity at 600 nm reached 1.0 and were then induced by 10 mM *L*-arabinose for 3 h. The bacterial cells were imaged using a Zeiss Axiovert fluorescence microscope.

### 2.7. Membrane Integrity Assay

To evaluate cell membrane integrity of MPAO1 with the overexpression of PftP4, the SYTOX™ green dead cell stain was used. Cultures were grown at 37 °C until the turbidity at 600 nm reached 1.0 and were then induced by 10 mM L-arabinose for 1 h, followed by staining with 200 nM of SYTOX™ green stain for 5 min. Then, bacterial cells were imaged using a Zeiss Axiovert fluorescence microscope (Carl Zeiss, Jena, Germany). The membrane permeability indicator SYTOX™ green stain is capable of penetrating into disrupted cell membrane but is unable to penetrate into the intact membrane of living cells.

### 2.8. Gene Knockout in P. aeruginosa MPAO1

Previous reported methods were used for the deletion of *PA0726* (*pftO*) and *pftp4* [31]. In detail, the upstream and downstream fragments of *pftP4* and *pftO* were amplified from the MPAO1 genome through PCR, using genomic DNA as the template using the primers listed in Table 2. Then, the purified PCR products were ligated into the EcoRI and HindIII pEX18Gm-modified suicide plasmid with Exnase II. Then, the recombination plasmids were transferred into MPAO1 strains via conjugation. In-frame deletion mutants were generated with the sucrose resistance selection method. The correct deletion mutants were further checked using both PCR and Sanger sequencing.

### 2.9. Phage Plaque Assay

Phages were isolated from MPAO1 Δ*pftP4* and MPAO1 Δ*pftO* overnight cultures using 0.22 μm microporous membranes. Then, 6 h cultures of MPAO1 and MAPO1 ΔPf4ΔPf6 were mixed 1:3 with melted R-Top medium and were then overlaid onto an LB medium plate. After tenfold serial dilutions of phage were spotted on the top, the double-layer plates were incubated at 37 °C overnight.

### 2.10. Quantitative PCR (qPCR)

Overnight cultures of MPAO1/pHERD20T, MAPO1/pHERD20T-*xisF4*, MPAO1 Δ*pftO*/pHERD20T, and MPAO1 Δ*pftO*/pHERD20T-*xisF4* were diluted 1/100 and cultured in LB medium at 37 °C. When the turbidity at 600 nm of cultures reached 1.0, 10 mM *L*-arabinose was added into the cultures. After 3 h of induction, DNA was collected from these bacterial samples. The number of total cells was quantified by qPCR using primers *gyrB*-qF/qR. The frequency of phage excision and replication in the bacteria was determined using primers *attB*-qF/R and *attP*-qF/R, respectively. Finally, the frequency of prophage excision was obtained as a ratio of *attB*/*gyrB*, while the frequency of prophage replication was obtained as a ratio of *attP*/*gyrB* using the calculation method that was described previously [32].

### 2.11. Quantitative Reverse Transcriptase PCR (qRT-PCR)

Overnight cultures of *E. coli* MG1655, *P. stuartii*, and *H. meridiana* 43005 with pHERD20T-*pftP4* or pHGE- *pftP4* were diluted 1/100 and were cultured until the turbidity at 600 nm reached 1.0. Then, 10 mM *L*-arabinose (for *E. coli* MG1655 and *P. stuartii*) or 1 mM IPTG (for *H. meridiana* 43005) was added into the cultures or not. After 3 h of induction, total RNA was extracted using the RNAprep Pure Cell/Bacteria Kit (Tiangen, Beijing, China). Then, 200 ng total RNA was used to generate cDNA using the HiScript II QRT SuperMix (Vazyme Biotech, Nanjing, China). Total cDNA was used for qPCR and the relative expression of *pftP4* was determined using primers *pftP4*-qF/qR.

## 3. Results

### 3.1. PftP4 (PA0726.1) Is a Toxic Polypeptide and pftO (PA0726) Is Responsible for Phage Secretion

The genomes of Pf phages consist of the core genes needed for phage replication and assembly, as well as accessory genes whose functions mostly remain unknown [29]. In the core region of the Pf4 prophage, downstream of the five genes (PA0721–PA0725) encoding the five capsid proteins for phage virion, is the *PA0726* gene. This gene encodes a putative Zona occludens toxin and was first characterized in the filamentous CTX phage as a virulence gene [1]. Following analysis, it is suggested that this protein is involved with the secretion of the Pf phage [33,34]. Interestingly, when the coding region was analyzed, we found a previously unannotated open reading region frame that shares the start codon (ATG) with the stop codon of the *PA0726* gene (TGA). The gene *PA0726.1* contains 102 bp, which could be translated into a 33-amino-acid polypeptide (Figure 1A).

To test the function of these genes, we first constructed deletion mutants by carefully removing most of the coding regions without interfering with the neighboring gene. We then overexpressed the Pf4 excisionase gene *xisF4*, which serves as the phage activator and induces the excision and replication of Pf4 in the planktonic cells of MPAO1. As expected, non-superinfective Pf4 was produced in the MPAO1 effluent when tested using MPAO1 and the prophage-deleted strain ΔPf4ΔPf6 (Figure 1B). Importantly, deleting *PA0726* completely blocks phage production in the effluent, indicating that PA0726 is critical for phage virion production. In contrast, deleting the *PA0726.1* gene did not affect Pf4 phage production (Figure 1C). We also employed qPCR to determine the copy number of the replicative form of Pf4 (RF Pf4) in the *PA0726* mutant. Deleting this gene did not affect the ability of *xisF4* to induce the replication of Pf4, further suggesting that PA0726 participated in phage secretion, and PA0726.1 did not have an impact on this process. We thus proposed naming PA0726 as PftO (Pf export to outside) and PA0726.1 as PftP (Pf-encoded toxic polypeptide).

To further study the function of PftP4, we cloned the full length of the *pftP4* gene into the pHERD20T plasmid to construct pHERD20T-*pftP4*. A toxicity assay and LIVE/DEAD™ *Bac*Light™ bacterial viability assay were conducted with and without the induction of 10 mM *L*-arabinose to produce PftP4. The results demonstrated the significant lethal effect of PftP4 on *P. aeruginosa* (Figure 1E,F). Additionally, as depicted in Figure 1A, the region coding *pftP4* also contains another two potential open reading frames (ORFs), each with putative ribosome binding sites (RBSs) and a putative start codon ATG, which could generate a 22-or 11-amino-acid-long peptide, respectively. Each of these sequences, referred to as PftP4-M (the last 22 amino acids in length) and PftP4-S (the last 11 amino acids in length), was individually cloned and transformed into MPAO1 (Figure 1A). The toxicity assay revealed that neither of the two shorter peptides exhibited toxicity to MPAO1 (Figure 1E). Considering that the first RBS is the strongest (GGAGG), it is likely that the predominant product of *pftP4* is the full-length 33-aa-long polypeptide.

### 3.2. PftP4 Is Comonly Found in the Inoviridae Family

To investigate the prevalence of PftP4 homologues in phages, we conducted a search for the PftP4 polypeptide sequences within the genomes of both cultivated and uncultivated viruses deposited in the IMG/VR database (https://genome.jgi.doe.gov/portal/IMG_VR/IMG_VR.home.html) (accessed on 11 January 2024) [35]. Our search yielded 427 PftP4 homologues with >80% identity, the majority of which belong to the *Inoviridae* family, with only one exception found in the *Caudoviricetes* family (Appendix A). Additionally, phages harboring this toxic polypeptide are from a diverse range of ecosystems, including those associated with free living in the environment, eukaryotic hosts, and engineered phages through synthesis or those used for wastewater treatment (Figure 2A). Furthermore, our host prediction revealed that these phages selectively infect *Pseudomonas*, particularly *P. aeruginosa* (Appendix A). Based on topology analysis, a large proportion of the identified phages are provirus (also called prophage), while a smaller fraction of phages possess linear genomes (Figure 2B). These findings suggest that these polypeptides are enriched in *Pseudomonas* filamentous prophages.

The Pf prophage is prevalent among *P. aeruginosa* isolates and it has been categorized as Pf1 to Pf8 [36]. Notably, PftP4 homologues were identified in phages Pf1, Pf4, Pf5, Pf6, Pf7, and Pf8 with a high sequence identity (Figure 2C). To investigate whether PftP4 homologues in other phages exerted toxicity, a toxicity assay was conducted for PftP derived from Pf1, Pf6, and Pf7 prophages (the amino acid sequences of PftP1, PftP5, and PftP8 are identical). The expression of full length PftP1/5/8, PftP6, and PftP7 also exerted significant toxicity on MPAO1 (Figure 2D,E), suggesting that the variations in amino acids among the Pf sequences did not affect their toxic properties. Consistent with PftP4, neither PftP6-M nor PftP6-S showed toxicity of PftP6 (Figure 2D). These findings suggest that despite the presence of several non-conserved amino acids among PftP4 homologues, their toxic characteristics towards *P. aeruginosa* remain conserved.

### 3.3. The Toxicity of PftP4 to P. aeruginosa Is Specific

To investigate whether PftP4 is toxic to different species of bacteria, we introduced *pftP4* into *E. coli* MG1655, *P. stuartii*, and *H. meridiana* 43005, and subsequently performed a toxicity assay. The results showed that PftP4 did not exhibit toxicity towards *E*. *coli* MG1655, *P*. *stuartii*, or *H*. *meridiana* 43005 (Figure 3A). To exclude that this is due to the low or lack of expression of PftP4 in these non-*P. aeruginosa* strains, qRT-PCR of *pftP4* transcripts was conducted. The results of qRT-PCR indicated that PftP4 was induced significantly in these non-*P. aeruginosa* strains with expression plasmids (Figure 3B). Thus, we hypothesized that PftP4 specifically exerted toxicity towards *P. aeruginosa*.

To further substantiate this hypothesis, we introduced PftP4 into different clinically isolated strains of *P. aeruginosa*, including WK172, PA-16, PA-19, PA-20, and PA-22. As shown in Figure 3A, PftP4 exerted toxicity towards both the standard laboratory strains and clinical isolates. These findings underscore the broad spectrum of toxic effects exerted by PftP4 on *P. aeruginosa*, highlighting its significant clinical relevance.

### 3.4. PftP4 Is Membrane-Anchored and Increases the Cell Membrane Permeability of P. aeruginosa

A transmembrane domain from L7 to P32 was predicted in PftP4 using the Transmembrane Hidden Markov Model (Figure 4A). Additionally, protein structure prediction revealed that the majority of PftP4 sequences formed a helical structure from W4 to V29 (Figure 4B), aiding the localization of PftP4 within the membrane. To experimentally validate the localization of PftP4 in MPAO1, a GFP-tag was fused to the N-terminus of PftP4 in pHERD20T-*gfp*-*pftP4*, and the MPAO1 cells with recombination plasmids were observed using a fluorescence microscope. As expected, GFP-PftP4 was observed to be localized at the cell poles, while a uniform green-fluorescent distribution was observed in cells expressing GFP as a negative control, indicating its status as a membrane-anchored polypeptide (Figure 4C).

Proteins with transmembrane domains play a role in various dynamic cellular processes, such as substance transport, signal transduction, cell recognition, and adhesion [37]. To explore the role of PftP4 in membrane integrity, SYTOX^TM^ green nucleic acid stain, a green-fluorescent dye impermeable to live cells but able to penetrate compromised membranes characteristic of dead cells, was used. A high level of green fluorescence observed in cells overexpressing *pftP4* represented the formation of pores within the inner membrane, and interestingly, the host DNA displayed a wavy shape (Figure 4D). As a control, using an empty vector, the majority of cell membranes were intact and even for a very small proportion of cells with (~1%) that have disrupted membrane still showed uniform DNA topology. These results suggested that PftP4 exerts its toxic activity on *P. aeruginosa* by disrupting cell membrane integrity and potentially altering the DNA topology of host genomic DNA.

## 4. Discussion

Filamentous phages are prevalent in both bacteria and archaea, possessing the simplest genome, and they harbor diverse genes that directly and significantly influence *P. aeruginosa* infections [23,24,25,33,38,39]. Pf and its encoding genes have been considered potential targets for treating *P. aeruginosa* [21,40,41,42]. However, there were few reports on their direct application as drugs to control *P. aeruginosa* infections. We intended to explore polypeptides derived from Pf phages that could eliminate *P. aeruginosa*. We discovered that the filamentous Pf prophages encoded a membrane-anchored polypeptide, PftP4, which specifically exhibited toxicity to *P. aeruginosa* among the tested bacterial strains. These findings highlighted the potential clinical significance of PftP4 in inhibiting the proliferation of the *P. aeruginosa* pathogen and its biosafety. Furthermore, the underlying mechanism of PftP4 toxicity to *P. aeruginosa* indicated that it may influence the growth of *P. aeruginosa* through disrupting its cell membrane and potentially altering the topology of the host genome simultaneously. The genomes of Pf phages and *P. aeruginosa* share certain similarity due to the lysogeny of Pf prophages and horizontal gene transfer, leading to the specific toxicity activity of PftP4 in *P. aeruginosa* rather than in other bacteria. This may be the outcome of a long-term co-evolution and an arms race between bacteriophages and their hosts.

PftO and PftP4 play distinct roles in terms of Pf4 production, and only PftO is involved in the release of Pf4. PftP4 exhibits strong toxicity towards host cells, and its overexpression during phage propagation significantly hinders the production of phage progeny. Interestingly, both genes are prominently induced to express in biofilms [20]. We previously showed that overexpressing phage capsid proteins PA0724 (encoded by *gIII*), PA0723 (encoded by *gVIII*), and PA0721 (encoded by *gVII*; also called *pfsE*) was also toxic [19,23,39], and the secretion of Pf to the outside mediated by PftO can thus alleviate the toxicity caused by the accumulation of these membrane-anchored capsid proteins. As concerns another aspect, PftP4 can serve as a brake to maintain phage production to a specific level through an abortive infection-like mechanism, as the production of an excess of PftP4 could lead to host cell death. Unlike Ff and M13 filamentous phages infecting *E. coli*, the infection of Pf phages including superinfective Pf4 and Pf6 can lead to host death and form lytic phage-like plaques on bacterial lawns. The presence of unique toxic polypeptides PftP4 and PftP6 in Pf phages might be responsible for the increased infectivity in these Pf phages.

Our investigation into the discovery of an efficient anti-*P. aeruginosa* peptide PftP4 has provided valuable candidates for *P. aeruginosa* treatments. Our aim was to externally administer PftP4 as an inhibitor of *P. aeruginosa*. To achieve this, we synthesized the polypeptide PftP4 (dissolved in DMSO) and utilized DMSO as a control to assess the impact of exogenous polypeptide PftP4 on the growth of MPAO1. Contrary to our assumptions, the addition of polypeptide PftP4 to the medium had minimal impact on the growth of MPAO1. This lack of impact may be attributed to the inability of the PftP4 polypeptide to effectively penetrate the cells from the external environment. It has been demonstrated that bacterial clustering and biofilm formation significantly compromise the efficacy of drugs [43]. Antimicrobial efficacy could potentially be enhanced through the use of targeted drug delivery systems with biocompatible properties [44,45]. Our findings have demonstrated the potential of PftP4 to inhibit the growth and induce the death of *P. aeruginosa* through overproduction. Therefore, it is pertinent to explore suitable drug delivery vehicles for PftP4. If the polypeptide PftP4 can be loaded onto delivery vehicles and effectively penetrate *P. aeruginosa*, it may efficiently eliminate the pathogenic bacteria *P. aeruginosa*, thereby safeguarding human health. This presents a promising and potentially effective approach for the complete and safe treatment of *P. aeruginosa* infections.

## 5. Conclusions

In this study, our collective findings demonstrate that PftP4 and its homologues, encoded by various Pf prophages in *P. aeruginosa,* exerted a potent and specific bactericidal effect on both standard model strains and clinical strains. This not only enhances our comprehension of the functions of the accessory genes of Pf phages in *P. aeruginosa* but also introduces a candidate approach for treating *P. aeruginosa* infections, holding significant clinical relevance. Future research will focus on delivering the PftP4 polypeptide into cells and exploring the application of PftP4 and its homologues in the treatment of *P. aeruginosa* within hospital environments. In addition, filamentous phages are present in a diverse array of bacterial and archaeal hosts [38], and their encoding peptides as potential therapeutic agents for bacterial infections need further investigation.

## Figures and Tables

**Figure 1 viruses-17-00112-f001:**
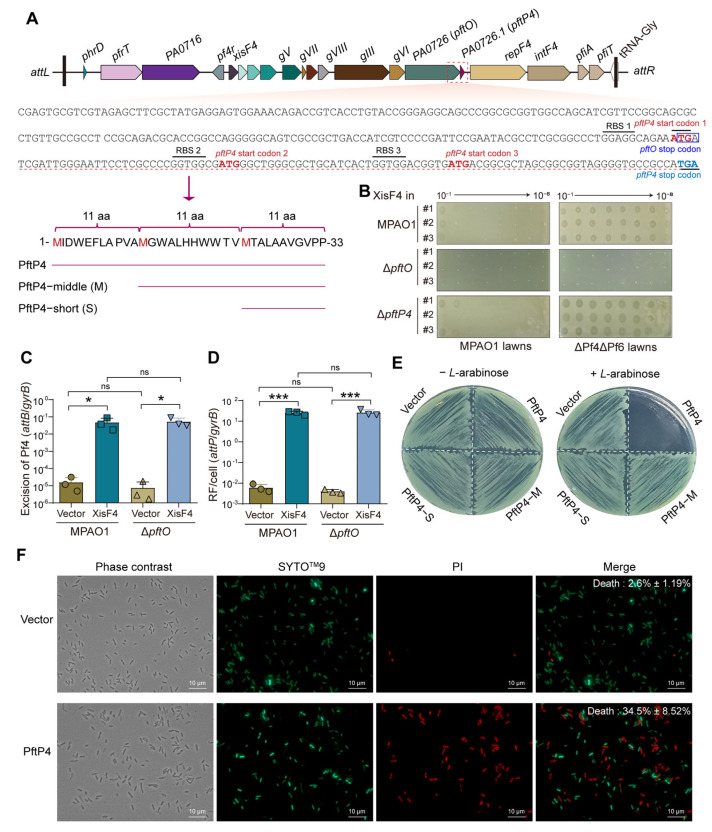
PftP4 in the Pf4 prophage is a toxic polypeptide. (**A**) Gene loci and related sequence analysis of *pftP4* in the Pf4 prophage. RBS: predicted ribosome binding site. (**B**) Pf4 phages were obtained from the planktonic culture supernatants of MPAO1, Δ*pftO*, and Δ*pftP4* overproducing XisF4 via p-*xisF4*. Phage titers were assessed on MPAO1 and ΔPf4ΔPf6 lawns. (**C**) Excision of Pf4 phages in planktonic MPAO1 and Δ*PA0726* cells with XisF4 was overproduced via the p-*xisF4* plasmid. ns: not significant. Ns indicates not significant. (**D**) RF of Pf4 phages in cells in B. (**E**) Toxicity of PftP4 in MPAO1 hosts. Vector: pHERD20T empty plasmid. (**F**), Live/Dead staining of MPAO1 cells with PftP4 overproduction. The percentages of live cells with *pftP4* overexpression via pHERD20T(p)-*pftP4.* The pHERD20T empty vector was used as a control without any gene expression. The percentages of dead cells were labeled on the merged images. The green cells are live, while the red cells are dead. Three independent cultures were used for statistical analysis, and only representative images were shown in (**B**,**E**,**F**). Two-sided paired Student’s *t*-Test was used for comparisons between two groups with 95% confidence intervals. *p* < 0.05 was considered statistically significant. (* *p* < 0.05; *** *p* < 0.001) is indicated in (**C**,**D**).

**Figure 2 viruses-17-00112-f002:**
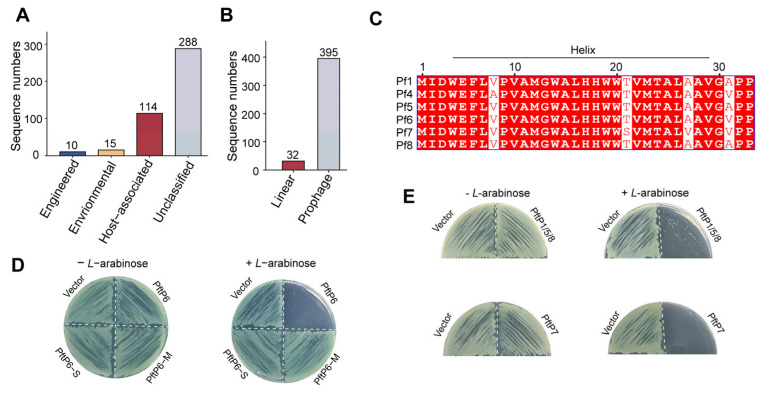
PftP4 is present in several *P. aeruginosa* prophages. (**A**) Ecosystem distribution of identified phages. (**B**) Topology of identified phages. (**C**) Sequence alignment of PftP4 homologues in Pf1, Pf4, Pf5, Pf6, Pf7, and Pf8 phages. (**D**) Toxicity of PftP6, PftP6-M, and PftP6-S in MPAO1 hosts. Vector: pHERD20T empty plasmid. (**E**) Toxicity of PftP1/5/8 and PftP7 in MPAO1 hosts. Three independent cultures were used for (**D**,**E**), and only representative images are shown.

**Figure 3 viruses-17-00112-f003:**
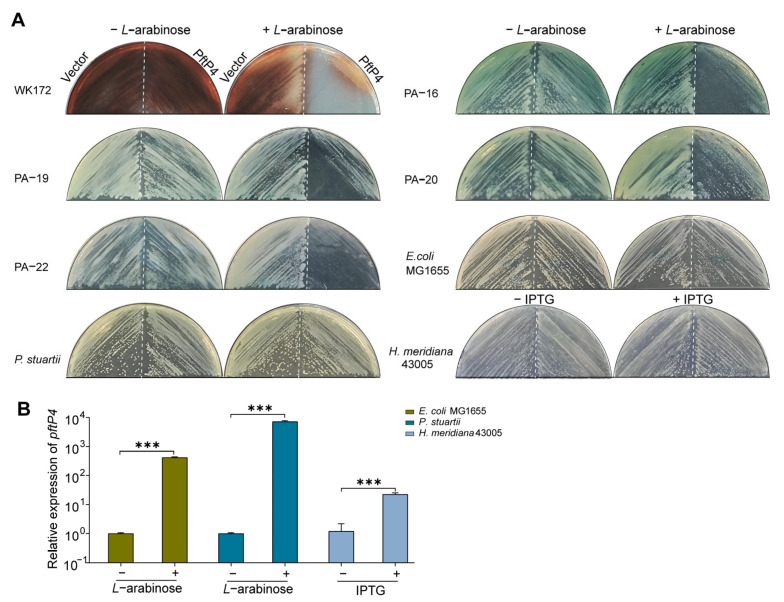
The toxicity of PftP4 is *P. aeruginosa*-specific. (**A**) Toxicity of PftP4 in different *P. aeruginosa* clinical strains, *E. coli* MG1655, *P. stuartii*, and *H. meridiana* 43005. The pHERD20T empty vector or the pHGE empty vector (only for *H. meridiana* 43005) was used as a control. In each plate, the left and right panels were streaked with the empty vector and *pftP4*-overexpressed cultures, respectively. Three independent replicates were used for each strain and only one representative image for each host is shown. (**B**) Relative expression of *pftP4* in *E. coli* MG1655, *P. stuartii*, and *H. meridiana* 43005 with *pftP4* being overexpressed or not. Three independent replicates were used for each strain. The relative expression of *pftP4* was normalized by its mRNA levels in cells without *pftP4* induction; data are shown as mean ± SD. Two-sided paired Student’s *t*-Test was used for comparisons between two groups with 95% confidence intervals. *p* < 0.001 was shown with three asterisks.

**Figure 4 viruses-17-00112-f004:**
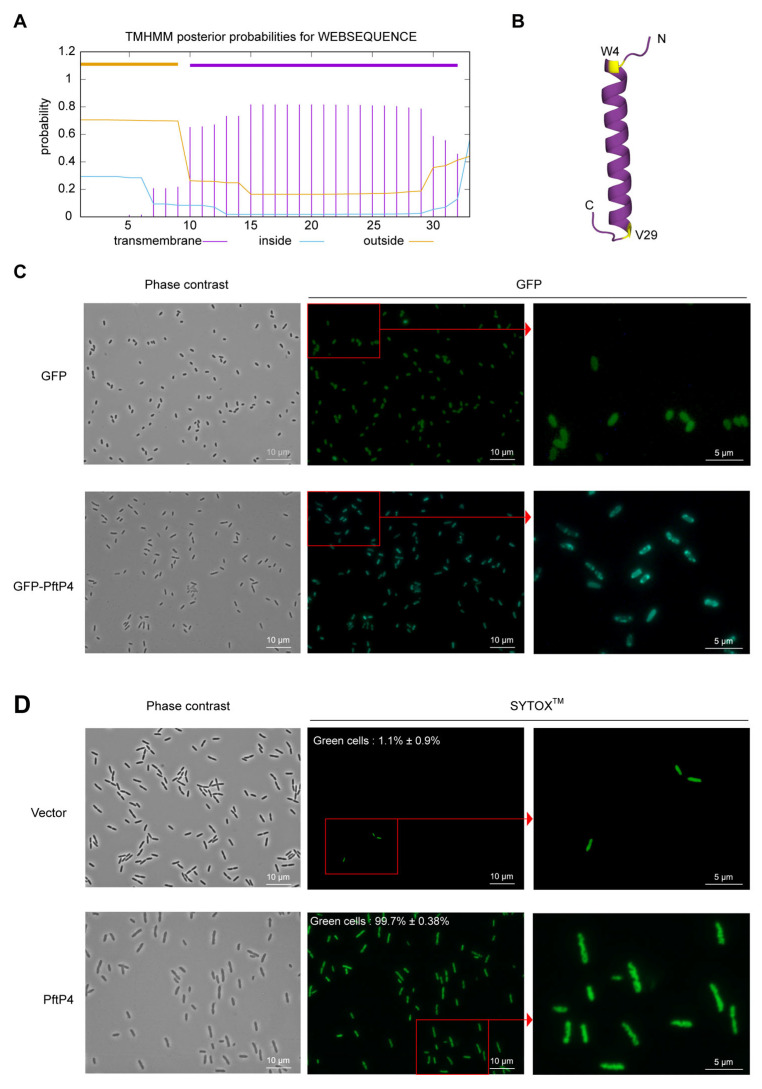
PftP4 was anchored in the cell membrane, and it increased the cell membrane permeability of MPAO1. (**A**) Transmembrane helices of the PftP4 polypeptide predicted using TMHMM. (**B**) The structure of PftP4 predicted with Alphafold2. The predicted structure is shown in violet purple, and the start and stop amino acid of the predicted helix are shown in yellow. (**C**) The subcellular localization of PftP4 in MPAO1. The location of green fluorescence represented the expression of the GFP tag. Three independent cultures were used for statistical analysis, and only representative images are shown. (**D**) Inner membrane permeability indicator SYTOX^TM^ staining of MPAO1 cells with PftP4 overproduction. The pHERD20T empty vector was used as a control without any gene expression. The percentages of cells with disrupted inner membranes were labeled on the SYTOX^TM^ fluorescence images. Three independent cultures were used for statistical analysis, and only representative images are shown.

**Table 1 viruses-17-00112-t001:** Strains and plasmids used in this study.

Strains/Plasmids	Description ^a^	Reference
*P. aeruginosa*		
MPAO1	Wild type	[20]
WK172	Clinical strains	This study
PA-16	Clinical *P. aeruginosa* (PA) strain	This study
PA-19	Clinical *P. aeruginosa* (PA) strain	This study
PA-20	Clinical *P. aeruginosa* (PA) strain	This study
PA-22	Clinical *P. aeruginosa* (PA) strain	This study
*E. coli*		
WM3064	*thrB*1004 *pro thi rpsL hsdS lacZ* ΔM15 RP4-1360 Δ(*araBAD*)567 Δ*dapA*1341::[*erm pir*(wt)]	W., Metcalf, UIUC
MG1655	F^−^ λ^−^ ilvG rfb-50 rph-1	[26]
*P.* *stuartii*	Clinical strains, isolated from hospital patients	This study
*H. meridiana* 43005	Isolated from coral samples of *G. fascicularis*.	[27]
Plasmids		
pHERD20T	*L*-arabinose-inducible expression vector, Car^R^	[24]
pHERD20T-*xisF4*	*xisF4* in pHERD20T NcoI/HindIII, Car^R^	[25]
pHERD20T-*pftP4*	*pftP4* in pHERD20T NcoI/HindIII, Car^R^	This study
pHERD20T-*pftP4*-M	*pftP4*-M in pHERD20T NcoI/HindIII, Car^R^	This study
pHERD20T-*pftP4*-S	*pftP4*-S in pHERD20T NcoI/HindIII, Car^R^	This study
pHERD20T-*pftP1*	*pftP1* in pHERD20T NcoI/HindIII, Car^R^	This study
pHERD20T-*pftP6*	*pftP6* in pHERD20T NcoI/HindIII, Car^R^	This study
pHERD20T-*pftP7*	*pftP7* in pHERD20T NcoI/HindIII, Car^R^	This study
pHERD20T-*pftP6*-M	*pftP6*-M in pHERD20T NcoI/HindIII, Car^R^	This study
pHERD20T-*pftP6*-S	*pftP6*-S in pHERD20T NcoI/HindIII, Car^R^	This study
pHERD20T-*gfp*-*pftP4*	*gfp*-*pftP4* in pHERD20T NcoI/HindIII, Car^R^	This study
pHGE	IPTG-inducible expression vector, Cm^R^	[28]
pHGE-*pftP4*	*pftP4* in pHGE EcoRI/HindIII, Cm^R^	This study
pEx18Gm	Gene replacement vector, Gm^R^, *oriT*^+^, *sacB*^+^	[29]
pEx18Gm-ud-*pftP4*	The upstream and downstream of *pftP4* in pEx18Gm EcoRI/HindIII, Gm^R^	This study
pEx18Gm-ud-*PA0726* (*pftO*)	The upstream and downstream of *PA0726* (*pftO*) in pEx18Gm EcoRI/HindIII, Gm^R^	This study

^a^ Car^R^: carbenicillin resistance; Cm^R^: chloramphenicol resistance; Gm^R^: Gentamicin.

**Table 2 viruses-17-00112-t002:** Primers used in this study.

Primers	Sequence (5′-3′)
pHERD20T-F	ATCGCAACTCTCTACTGTTTCT
pHERD20T-R	TGCAAGGCGATTAAGTTGGGT
pHGE-F	CACCTCGCTAACGGATTCACC
pHGE-R	CCAATACGCAAACCGCCTC
*pftP4*-F	AGGAGATATACATACCCATGATCG ATTGGGAATTCCTCGCCCCGGTGGCG
*pftP4*-R	ACGACGGCCAGTGCCAAGCTTTCATGGCGGC ACCCCTACCGCCGCTAGCGCCGTCATCACCGT
*pftP1*-F	AGGAGATATACATACCCATGA TCGATTGGGAATTCCTCGTTC
*pftP1*-R	ACGACGGCCAGTGCCAAGCTTTC ATGGCGGCGCCCCTACCGCCGCT
*pftP6*-F	AGGAGATATACATACCCATGATCGATT GGGAATTCCTCGTTCCGGTGGCG
*pftP6*-R	ACGACGGCCAGTGCCAAGCTTTCATGGCGGC ACCCCTACCGCCGCTAGCGCCGTCATCACCGA
*pftP7*-F	AGGAGATATACATACCCATGATCGATTGGGAATTC CTCGTCCCGGTGGCGATGGGCTGGGCGCTGCATCA CTGGTGGTCGGTGATGACGGCGCTAGTGGCGGTAG GGGTGCCGCCATGAAAGCTTGGCACTGGCCGTCGT
*pftP7*-R	ACGACGGCCAGTGCCAAGCTTTCATGGCGGCACCC CTACCGCCACTAGCGCCGTCATCACCGACCACCAG TGATGCAGCGCCCAGCCCATCGCCACCGGGACGAG GAATTCCCAATCGATCATGGGTATGTATATCTCCT
*pftP4*-M-F	AGGAGATATACATACCCATGGGCTGGGCGCT GCATCACTGGTGGACGGTGATGACGGCGCTA GCGGCGGTAGGGGTGCCGCCATGAAAGCTTG GCACTGGCCGTCGT
*pftP4*-M-R	ACGACGGCCAGTGCCAAGCTTTCATGGCGGCACCC CTACCGCCGCTAGCGCCGTCATCACCGTCCACCAGT GATGCAGCGCCCAGCCCATGGGTATGTATATCTCCT
*pftP4*-S-F	AGGAGATATACATACCCATGACGGCGCTAGCGGCGG TAGGGGTGCCGCCATGAAAGCTTGGCACTGGCCGTCGT
*pftP4*-S-R	ACGACGGCCAGTGCCAAGCTTTCATGGCGGCACCCCT ACCGCCGCTAGCGCCGTCATGGGTATGTATATCTCCT
*pftP6*-M-F	AGGAGATATACATACCCATGGGCTGGGCGCTGCATCA CTGGTGGACGGTGATGACGGCGCTAGCGGCGGTAGGG GTGCCGCCATGAAAGCTTGGCACTGGCCGTCGT
*pftP6*-M-R	ACGACGGCCAGTGCCAAGCTTTCATGGCGGCACCCCT ACCGCCGCTAGCGCCGTCATCACCGTCCACCAGTGAT GCAGCGCCCAGCCCATGGGTATGTATATCTCCT
NcoI-*gfp*-F	AGGAGATATACATACCCATGAG TAAAGGAGAAGAACTTTT
*gfp*-*pftP4*-R	CGAGGAATTCCCAATCGATTTTGTA TAGTTCATCCATGC
*gfp*-*pftP4*-F	GCATGGATGAACTATACAAAATCGATT GGGAATTCCTCGCCCCGG
*pftP4*-HindIII-R	ACGACGGCCAGTGCCAAGCTTTCATGGCGGC ACCCCTACCGCCGCTAGCGCCGTCATCACCGT
pHGE-*pftP4*-EcoRI-F	ATTTCACACAGGAGAGAATTATGATCGATTGGGAATTCCTCGCCCCGGTGGCG
pHGE-*pftP4*-HindIII-R	CCGCCAAAACAGCCAAGCTTTCATGGCGGCACCCCTACCGCCGCTAGCGCCGTCATCACCGT
*pftP4*-up-F	ACGACGGCCAGTGCCAAGCTTGCTGGTTCTTCTACTGGCAT
*pftP4*-up-R	CCGGCGGCGCGGCCCGCCCCTCATTTCTGCCTCCAGGGCC
*pftP4*-down-F	GGCCCTGGAGGCAGAAATGAGGGGCGGGCCGCGCCGCCGG
*pftP4*-down-R	TATGACCATGATTACGAATTCTCAGACCCATTTCAGCGTTC
*pftP4*-LF	ATAACGGCGGCGGCAATGAC
*pftP4*-LR	ACACGCTTTCCCAATAGTCG
*pftO*-up-F	ACGACGGCCAGTGCCAAGCTTACGCCAAGCAAGAACTCAAG
*pftO*-up-R	AGGAATTCCCAATCGATCATGGATCACCTCCCAATGAACG
*pftO*-down-F	CGTTCATTGGGAGGTGATCCATGATCGATTGGGAATTCCT
*pftO*-down-R	TATGACCATGATTACGAATTCTCAGACCCATTTCAGCGTTC
*pftO*-LF	TGGCGGTGGAAACAATAACG
*pftO*-LR	TTGGTTGGTTTCGCAGTGAC
*gyrB*-qF	CAAGTACGAAGGCGGTCTGAAG
*gyrB*-qR	GCAGAGCAGGTTCTCGTTGAA
*attB*-qF	CGCGGTAAAGCCTGAAAACA
*attB*-qR	GCCGCGCATCTGGTAAAA
*attP*-qF	GGGCTTGGCAGGGTGATT
*attP*-qR	TCATGCAGGACCTGTCGAAA
*pftP4*-qF	ATCGATTGGGAATTCCTCGC
*pftP4*-qR	ATGGCGGCACCCCTAC

## Data Availability

All the supporting data are available in the paper and Appendix A.

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
