# Peer review of "Inovirus-Encoded Peptides Induce Specific Toxicity in Pseudomonas aeruginosa"

_viruses, 2025, doi:10.3390/v17010112_

Round 1
Reviewer 1 Report
Comments and Suggestions for Authors
The manuscript contributes valuable insights into the role of Pf prophages and their encoded peptides in bacterial pathogenesis and offers a promising candidate peptide for future therapeutic development. However, its impact is diminished by speculative conclusions.
Major Comment:
I believe there is a discrepancy between the results and the conclusions presented in the manuscript. For example, in Lines 19–20, it is stated that PftP4 decreases membrane permeability:
"Overexpression of PftP4 demonstrated cellular toxicity, with subcellular localization indicating its presence in the cell membrane and a subsequent reduction in membrane permeability." A similar statement is found in Lines 78–79: "We observed that PftP4 is a toxic polypeptide located in the membrane and it reduces membrane permeability". At the same time, other data indicate that PftP4 increases membrane permeability. This conclusion can be drawn from the data presented in Figure 4D, as well as from Lines 285 and 309, where it is specifically stated: "PftP4 is membrane anchored and increases cell membrane permeability of P. aeruginosa" and "PftP4 was anchored in the cell membrane, and it increased cell membrane permeability of MPAO1". Why does the manuscript claim in some sections that the peptide decreases membrane permeability, while in others, it claims the opposite? Please verify and clarify this inconsistency.
Minor Comments:
Lines 70-71 and Lines 181-182. “Zona occludens toxin” and “Zonula occludens toxin” ─ Please clarify which term is correct and ensure consistency throughout the manuscript.
Line 71. “PA0728 encodes the integrase [25]” ─ I could not find a mention of this in the cited reference #25. Please confirm the relevance of this citation or the accuracy of the statement.
Author Response
Reviewer 1
The manuscript contributes valuable insights into the role of Pf prophages and their encoded peptides in bacterial pathogenesis and offers a promising candidate peptide for future therapeutic development. However, its impact is diminished by speculative conclusions.
Major Comment:
I believe there is a discrepancy between the results and the conclusions presented in the manuscript. For example, in Lines 19–20, it is stated that PftP4 decreases membrane permeability:
"Overexpression of PftP4 demonstrated cellular toxicity, with subcellular localization indicating its presence in the cell membrane and a subsequent reduction in membrane permeability." A similar statement is found in Lines 78–79: "We observed that PftP4 is a toxic polypeptide located in the membrane and it reduces membrane permeability". At the same time, other data indicate that PftP4 increases membrane permeability. This conclusion can be drawn from the data presented in Figure 4D, as well as from Lines 285 and 309, where it is specifically stated: "PftP4 is membrane anchored and increases cell membrane permeability of P. aeruginosa" and "PftP4 was anchored in the cell membrane, and it increased cell membrane permeability of MPAO1". Why does the manuscript claim in some sections that the peptide decreases membrane permeability, while in others, it claims the opposite? Please verify and clarify this inconsistency.
Response: We apologize for the inconsistent statement in the manuscript. The description “PftP4 could increase cell membrane permeability of P. aeruginosa” is correct. We have revised other statements across the manuscript. Please see lines 22 and 83.
Minor Comments:
Lines 70-71 and Lines 181-182. “Zona occludens toxin” and “Zonula occludens toxin” ─ Please clarify which term is correct and ensure consistency throughout the manuscript.
Response: Thank you. The reference to “Zona occludens toxin” in Lines 70-71 is correct. We have rectified the mention in Lines 187-188. Additionally, we have reviewed the entire manuscript as advised.
Line 71. “PA0728 encodes the integrase [25]” ─ I could not find a mention of this in the cited reference #25. Please confirm the relevance of this citation or the accuracy of the statement.
Response: Thank you. We have updated the reference to another paper where PA0728 was identified as an integrase (Ref: Excisionase in Pf filamentous prophage controls lysis-lysogeny decision-making in Pseudomonas aeruginosa, 2019, 111, (2), 495-513).
Reviewer 2 Report
Comments and Suggestions for Authors
The author's manuscript satisfied all requirements for successful publication.
Author Response
Reviewer 2
The author's manuscript satisfied all requirements for successful publication.
Response: Thank you for your positive comments.
Round 2
Reviewer 1 Report
Comments and Suggestions for Authors
I have reviewed the revised manuscript submitted by the authors and believe this version is suitable for publication. The authors have addressed all the comments, and I can recommend the manuscript for publication in its current form.